# Weight Status Prediction Using a Neuron Network Based on Individual and Behavioral Data

**DOI:** 10.3390/healthcare11081101

**Published:** 2023-04-12

**Authors:** Sylvie Rousset, Aymeric Angelo, Toufik Hamadouche, Philippe Lacomme

**Affiliations:** 1University Clermont Auvergne, UNH, UMR1019, INRAE, 63000 Clermont Ferrand, France; aymeric.angelo@gmail.com (A.A.); toufik.hamadouche@etu.uca.fr (T.H.); 2University Clermont Auvergne, LIMOS UMR CNRS 6158, 63000 Clermont Ferrand, France; placomme@isima.fr

**Keywords:** weight status, physical activity, diet, neural network, prediction, classification, supervised learning

## Abstract

Background: The worldwide epidemic of weight gain and obesity is increasing in response to the evolution of lifestyles. Our aim is to provide a new predictive method for current and future weight status estimation based on individual and behavioral characteristics. Methods: The data of 273 normal (NW), overweight (OW) and obese (OB) subjects were assigned either to the training or to the test sample. The multi-layer perceptron classifier (MLP) classified the data into one of the three weight statuses (NW, OW, OB), and the classification model accuracy was determined using the test dataset and the confusion matrix. Results: On the basis of age, height, light-intensity physical activity and the daily number of vegetable portions consumed, the multi-layer perceptron classifier achieved 75.8% accuracy with 90.3% for NW, 34.2% for OW and 66.7% for OB. The NW and OW subjects showed the highest and the lowest number of true positives, respectively. The OW subjects were very often confused with NW. The OB subjects were confused with OW or NW 16.6% of the time. Conclusions: To increase the accuracy of the classification, a greater number of data and/or variables are needed.

## 1. Introduction

The prevalence of overweight and obesity is increasing in France, as it is worldwide. In 2020 [1], only 52.7% of the French adult population was normal weight, whereas 17.0% was obese and 30.3% was overweight. In 2030, Ward et al. (2019) predict that almost 50% of the American population will be obese [2], and, in France, the projection of obesity prevalence in 2060 is 25% [3]. Overweightness occurs following an energy imbalance: the individual consumes more energy than he/she expends. Lifestyles have dramatically changed over the last decades. An increasingly sedentary lifestyle coupled with unhealthy dietary practices are key risk factors for obesity [4]. The lack of physical activity and/or a fat- and sugar-rich diet are inducer factors of weight gain. Individual characteristics such as age and epoch play a determinant role in body mass index (BMI) and fat mass [5]. Santos et al. (2020) showed a progressive increase in BMI up to the age of 60, followed by a decrease in later years up to age 90 in a cohort of 50,019 Spanish people [6]. BMI was lower in younger people than in older ones. Among the 10,041 participants who had a normal BMI at baseline, 30% became overweight and 3.5% obese ten years later [6]. Moreover, the period also impacts BMI and weight status. Compared to 2007, the mean BMI of 18–30-year-olds in 2017 was about 2 kg.m^−2^ higher [6]. Staub et al. (2016) showed that the weight and BMI of Swiss conscripts increased in two distinct steps: at the end of the 1980s and again after 2002. Since 2010, the BMI has stabilized at a high level [7]. Nutritional and/or physical activity patterns may have changed during those periods. Thus, in the late 1980s, an increasing number of fast-food restaurants appeared in Switzerland and a marked price reduction of fuel made motorized transport easier. Consequently, a cheaper palatable and energy-rich diet coupled with reduced physical effort was made easily available.

Some authors have tried to model BMI based on food consumption and physical activity. Cundiff and Raghuvanshi (2012) showed that food group profiles and exercise levels determined the mean adult BMI independently of sex and the country’s gross domestic product [8]. For female and male cohorts, the mean BMI was positively correlated with alcohol, carbohydrates and total fat, and negatively with dietary fibers, polyunsaturated fatty acids and physical exercise. Some authors such as Hall (2009) worked to predict metabolic adaptation of body weight changes in humans. Models of simulation of body weight changes dependent on diet perturbation were shown [9]. An imbalance between the intake and selection of the macronutrients results in a change in body weight and composition. Another modeling simulated future trajectories of obesity in the English population from the prevalence of weight status (underweight, healthy weight, overweight, obese and very obese) observed between 1990 and 2000 [10]. The results of the linear and non-linear models depend on the assumptions made by the authors: the slowdown in BMI increase or the progressive increase in BMI over time. Even under the first hypothesis, the modeling predicts that the majority of the English population is at increased risk of becoming overweight or obese until at least 2035. In another model with a wave pattern for prevalence, obesity is expected to reach maximum levels between 2030 and 2052, according to the country [3].

Other observational studies are required to determine what types of physical activity and food products are concerned in the normal (NW), overweight (OW) and obese (OB) weight statuses. We hypothesized that being young, having significant physical activity and a healthy diet rich in plant products and low in snacks and fatty-salty-sugary products could be associated with normal weight status. The older the subjects, the less active they are and the less healthy their diet is, the higher their weight status is likely to be.

The aim of the present study is to predict the weight status (NW, OW and OB) of adults in free-living conditions on the basis of their individual characteristics and behavioral components using the Machine learning approach. It contributes to the understanding of the main determinants of weight status evolution over time and to the development of a mathematical model of prediction. A valorization of this work is to implement this model in a mobile application to inform people about the impact of behaviors on weight status over the years. This work is a step toward a new prevention system and is a contribution to the fight against obesity.

## 2. Materials and Methods

### 2.1. Subjects

This statistical study was carried out on existing data collected by a sample of the French population through their smartphone. This sample is representative of the French population for the proportion of the three major weight statuses: normal-weight, overweight and obese. 

This study was conducted on 273 data sets. The subjects were adults (older than 18 years), owned an Android smartphone and provided consent to the statistical processing of their data.

A total of 170 women and 103 men, either normal weight (NW, n = 139), overweight (OW, n = 74) or obese (n = 60), were studied in free-living conditions (Table 1). A total of 108 subjects were used for model development (constituting a training group consisting of 36 normal weight, 36 overweight and 36 obese subjects), and a total of 165 subjects (constituting a test group consisting of 103 normal weight, 38 overweight and 24 obese subjects) were used to evaluate the validity of the weight status estimation. 

### 2.2. Database

The data come from an existing collection. As a reminder, the WellBeNet application is presented below. the WellBeNet app [11] can be downloaded at the Play Store. This application has two parts to evaluate sedentary and active behaviors and food choices: eMouve and NutriQuantic. The time spent in four activity categories (immobility, light-, moderate- and vigorous-intensity activity) is expressed in percentages of the total recording time. eMouve provides an accurate estimation of time spent in the four categories, i.e., with errors in absolute value between 0.6% and 6.6%, and between 0.1% and 5.3%, respectively, in normal-weight and overweight subjects [12,13]. 

NutriQuantic is used to collect the number of portions consumed in 12 food categories [14].

### 2.3. Ethical Approval 

This study was conducted according to the guidelines laid down in the Declaration of Helsinki and the French legislation for the statistical treatments of existing anonymous human data. Written or verbal informed consent was obtained from all subjects for the aggregated treatment of their data. Verbal consent was witnessed and formally recorded. No specific ethical approval was necessary for this statistical study.

### 2.4. Statistical Models

The statistical analyses were performed using the SAS software version 9.4: normality tests, descriptive statistics, analyses of variance and mean comparison tests, analyses of correlation coefficients (Pearson). For the implementation of the neural network, Visual Studio (development environment), Python (computer language) and Scikit-Learn (main library of tools dedicated to learning) were used to find the best predictions of the three weight statuses. All of the variables studied in the training group were tested for normality using the Shapiro-Wilk test. For normally distributed variables, differences between weight statuses (normal weight, overweight and obese) were evaluated by analyses of variance (GLM procedure of SAS). Variations in other non-normally distributed variables were examined using Chochran-Mantel-Haenszel statistics (cmh, based on the difference in row mean ranks; FREQ procedure of SAS). A significant *p*-value (<0.05) indicates that the association between weight status and variable value is strong. When differences between the three statuses were significant, multiple mean comparison tests were performed (LSMEANS) on normally distributed variables. In non-normally distributed variables, the differences between NW and OW or between OW and OB were examined using Chochran-Mantel-Haenszel statistics. Correlation coefficients (Pearson) were then calculated between BMI and physical activity and food variables from the data collected by all the subjects in the training group. These preliminary statistical treatments provided information on variables relevant to the neural network. The neural network, multi-layer perceptron classifier (NN-MLP), was then used to learn the weight statuses among NW, OW and OB subjects based on individual and behavioral data. Different settings (number of neurons, number of epochs) were investigated in order to find the best configuration, i.e., the configuration that provides the highest number of correct classifications. The configuration was tested with two hidden layers. The number of neurons on the first layer varied from 100 to 250 in increments of 2. The second layer was composed of 20 to 100 neurons in increments of 10. Random weight was tested between 1 and 70. The number of epochs varied from 1 to 50. All the variables by groups of four to seven were introduced in the NN-MLP. All data collected by the subjects in the test group were used to validate this learning model. 

## 3. Results

### 3.1. Variables Discriminated by Weight Status

The relationships between the individual and behavioral characteristics and BMI category of the subjects were analyzed. Table 2 shows the most discriminant individual or behavioral variables among the 21 variables (age, height, sedentary behavior, light-, moderate-, vigorous-intensity activity, the total number of food portions and portions of nuts, vegetables, legumes, alcohol, hot drinks, fruit, dairy products, meat-fish-egg, refined and whole starch products, fatty-salty-sugary products and snacks) collected by the WellBeNet application. 

Seven variables were significantly discriminated by the weight status (NW, OW and OB): age (cmh = 8.3, *p* = 0.01), sedentary behavior (cmh = 40.6, *p* < 0.0001), light-intensity activity (F = 40.9, *p* < 0.0001), the total number of daily food portions (F = 2.4, *p* = 0.09), of nuts (cmh = 7.2, *p* = 0.03), vegetables (cmh = 5.69, *p* = 0.05) and legumes (cmh = 4.97, *p* = 0.08). The obese subjects were older, more sedentary and less active than both NW and OW subjects. They consumed fewer vegetables, legumes and nuts than NW subjects. The overweight subjects were the same age as NW subjects but they were more sedentary and less active and consumed fewer vegetables than NW subjects. 

For the other variables, no significant difference was observed between the three weight statuses.

### 3.2. Variables Linked to Body Mass Index

Linear dependence relationships between BMI values and values of individual and behavioral variables were searched. Table 3 shows the highest correlation coefficients between BMI and individual or behavioral variables recorded in this study. BMI and sedentary behavior were significantly positively correlated, while BMI and light- or moderate-intensity activity were negatively correlated. Thus, subjects with high BMI were less physically active and spent more time sitting than NW subjects. The total daily number of food portions, alcohol, fatty-salty-sugary products and vegetables were also negatively associated with BMI. That means that subjects with lower BMI reported a higher number of food portions than overweight subjects. Body mass index was positively correlated with age and negatively with height. Older and smaller subjects had higher BMIs.

### 3.3. Accuracy of the Neural Network to Classify the Subjects in the Three Weight Statuses

We tested a variety of neural network configurations to find the best predictions of the three weight statuses. The best configuration contained two layers with 120 and 20 neurons, 50 epochs and four input variables: age, height, amount of light-intensity activity and number of daily vegetable portions. Table 4 shows the confusion matrix of the training and test samples. In the test sample, NW was the most accurately predicted status with a rate of true positives of 93.2%, followed by OB with a rate of 66.7% and, finally, OW with 34.2%. The overall accuracy is 75.7%. The normal weight subjects were rarely confused with OW (4.9%) or with OB (1.9%). The overweight subjects were often confused with NW (52.6%) and sometimes with OB (13.2%). The obese subjects were confused as much with OW as with NW (16.6%).

### 3.4. Reasons for Confusion

We found many confusions and looked for the variables responsible for these confusions. Table 5 shows the mean characteristics for correctly and incorrectly predicted values in each weight status. The normal-weight subjects could be confused with OW or OB when light-intensity physical activity was low or when subjects were small in size or older in age than the average. The overweight subjects could be confused with NW when subjects were relatively young, active and tall. The overweight subjects could be confused with OB when subjects had very little physical activity, consumed few vegetables and were small. The obese subjects could be confused with OW when subjects were relatively young, moderately active and smaller than average. The obese subjects could be confused with NW when subjects were highly active and smaller than average. 

## 4. Discussion

The neural network MLP classifier found an average of 75.7% of true positives in the three weight statuses (NW, OW and OB) on the basis of four variables: two dealing with individual characteristics (age and height) and two with behavioral components (amount of light-intensity activity and daily consumption of vegetables). These findings are in agreement with other studies and lead us to consider or confirm that adherence to a healthy lifestyle that includes physical activity practiced regularly and a diet high in vegetables and fruits reduces the risk of obesity [15]. Researchers found that fatty food, alcohol and minimal physical activity increased the risk of weight gain, overweight and obesity, whereas walking, aerobic physical activity, consumption of foods containing dietary fibers, a Mediterranean diet and having been breastfed decreased this risk [16,17]. 

The NW status was more frequent in young subjects and was associated with a relatively high amount of light-intensity physical activity and of vegetable consumption compared to the OB status. Other studies have found that BMI is more frequently normal in the young population than in the elderly. Santos et al. (2020) studied a 10-year evolution of BMI in the same 50,019 participants aged 18 to 90 years. In both periods (2007–2008 and 2017–2018), the curve of BMI gradually increases until the age of 60 and then decreases between 60 and 90 years of age [6]. The mean BMI values varied from 23 to 30 kg/m^2^ and 26 to 30 kg/m^2^ between 18 and 60 years of age during both periods. Fat mass increases with age. Moreover, Han et al. (2016) showed an age-related loss of skeletal mass associated with lower muscle strength and poorer physical performance [18]. High-fat mass leads to fat infiltration into muscle. This type of infiltration contributes to decreased muscle quality and work performance [19]. Fat mass increases when energy brought by the diet becomes higher than energy expenditure. Between the ages of 25 and 54 people move less and have a diminished metabolic rate but maintain the same energy intake [20,21]. 

The normal weight subjects were infrequently mistaken for OW or OB subjects because of their non-specific characteristics but also because of their variability represented by large standard deviations. Thus, in this status, the subject could be older or younger, taller or shorter, more or less sedentary or active, or eat more or less vegetables. This variability in behaviors could correspond to more or less healthy metabolic statuses. Inconsistent metabolic status among people with the same normal-weight status was observed in scientific articles [22]. Since we did not characterize the metabolic status of the subjects, they may not all have been healthy even if their weight status was normal. These behaviors and individual characteristics may be sufficient to identify NW status but not to ensure good health over the long term.

The OW status was also characterized by a large variability of individual and behavioral characteristics. This status included less active and more sedentary and slightly older subjects than those with NW status. Small gaps appeared between NW and OW subjects. For this reason, OW subjects were often mistaken for NW subjects. These mistakes occurred when the age was similar to that of those with NW status and when the amount of light-intensity physical activity was comparable to that of NW subjects. Moreover, their BMI was lower than that of the rest of the overweight group and therefore closer to that of NW subjects. It is well known that some individuals are more likely to gain weight than others even if their behaviors are similar. Four metabolic factors were identified as being predictive of weight gain or obesity: low metabolic rate, low physical activity, low sympathetic nervous system activity and low fat oxidation [23]. Subjects with some of these metabolic characteristics and levels of physical activity comparable to that of NW status may nevertheless be overweight. Thus, not only behaviors but differences in metabolic rate as well can play a role in weight status. Less often, OW was mistaken for OB. It happened when subjects were older than average and when their light-intensity physical activity and vegetable consumption were low. Low consumption of vegetables and high consumption of fast foods and sweetened beverages in young adults were associated with weight gain and obesity [24]. The protective effect of vegetable consumption was demonstrated against weight gain [25]. In the present study, the overweight low consumers of vegetables had a higher BMI than the mean of the OW group and were, therefore, closer to OB. Differences in metabolic rate (resistance to weight gain) could also explain this confusion of statuses. 

The OB status was more often associated with increasing age [26]. When subjects age, unhealthy lifestyles during an extended period of time have deleterious consequences on both metabolic rate and body mass [27]. The OB status was sometimes confused with OW when subjects were younger and smaller, or confused with NW when they were twice more active than the OB average, as active as NW and smaller than the average. Other studies showed that various types of cardiorespiratory fitness were found in obese people [28]. Those who had higher VO2 max had healthier metabolic status [29]. These previous studies that integrate weight and metabolic parameters showed intermediate health statuses in obese people. As mentioned above, in addition to behaviors, metabolic rate can influence body mass. 

The first limitation of the present study concerns the low number of subjects, and the second is the measurement of BMI. Some subjects, especially those with borderline BMI based on the cut-off values (between normal weight and overweight, or between overweight and obese) may not be adequately classified. As weight was auto-reported, it could be underestimated. Underestimation of weight is common, especially in overweight and obese people [30,31]. This bias explained a part of the misclassified subjects, especially among those whose BMI is close to overweight or obese status. Measurement of weight and height by a health professional could improve weight status ranking. Another limitation is the lack of metabolism measurements (such as insulin resistance level, waist circumference, blood pressure and LDL cholesterol level) to discriminate the metabolic status in addition to BMI.

## 5. Conclusions

The prediction of weight status was accurate in about 76% of cases: better in normal weight (90%) and in obese (67%) than in overweight subjects (34%). Our findings show that OW was more often confused with normal weight than obese status. Being overweight is the result of an energy imbalance over time and weight gain increasing with age. It lacks variables and/or longitudinal data to better characterize it.

Since eating and physical behaviors potentially influence both metabolism and weight (body mass and composition), future studies could propose the introduction of metabolic parameters in the neural network to determine intermediate statuses and avoid confusion among them: healthy metabolic normal weight, unhealthy metabolic normal weight; healthy metabolic overweight, unhealthy metabolic overweight, healthy metabolic obese and unhealthy metabolic obese. 

Note that because age is one of the variables used by the neural network, it is possible to propose a simulation of weight status estimation 5 or 10 years later. Subjects can age themselves and become aware of whether they will maintain their weight status or not by keeping the same lifestyle habits. Moreover, they may realize that increasing light-intensity physical activity and/or the number of vegetable portions could prevent him/her from falling into the obese status. Thus, the application can be used as a prevention tool by making people aware of the long-term consequences of their behavior.

## Figures and Tables

**Table 1 healthcare-11-01101-t001:** Characteristics of both training and test groups. (Mean values and standard deviations (SD)).

	Training	Test
Variable	Mean (SD)	Mean (SD)
Sex (% women)	60.2	63.6
Weight status (% NW)	33.3	62.4
Weight status (% OW)	33.3	23.0
Weight status (% OB)	33.3	14.6
Age (years)	40.7 (11.2)	35.9 (12.9)
Height (m)	167.4 (9.2)	169.4 (10.0)
**NW**		
Weight (kg)	65.1 (9.1)	63.2 (8.4)
BMI (kg/m^2^)	22.5 (1.6)	21.7 (1.5)
**OW**		
Weight (kg)	76.6 (9.9)	78.9 (9.5)
BMI (kg/m^2^)	27.1 (1.4)	27.4 (1.3)
**OB**		
Weight (kg)	102.5 (18.2)	97.7 (14.3)
BMI (kg/m^2^)	36.9 (7.0)	35.6 (4.5)

NW: normal weight; OW: overweight; OB: obese.

**Table 2 healthcare-11-01101-t002:** Mean comparison between the three weight statuses (NW, OW and OB; n = 108).

Variables	NW	OW	OB	*p*-Value
Age (y)	39.9 (11.5) ^a^	37.3 (11.2) ^a^	44.8 (9.66) ^b^	0.01
Sedentary behavior (%)	71.9 (11.1) ^a^	81.9 (8.0) ^b^	87.9 (7.1) ^c^	<0.0001
Light activity (%)	21.6 (10.2) ^a^	11.9 (6.9) ^b^	7.0 (5.4) ^c^	<0.0001
Total food portion (Nb)	13.0 (4.2) ^a^	12.5 (4.4) ^ab^	10.9 (4.1) ^b^	0.09
Nut portion (Nb)	0.48 (0.44) ^a^	0.33 (0.62) ^b^	0.27 (0.42) ^b^	0.04
Vegetable portion (Nb)	2.06 (1.40) ^a^	1.43 (0.74) ^b^	1.50 (0.88) ^b^	0.02
Legume portion (Nb)	0.33 (0.85) ^a^	0.28 (0.49) ^a^	0.09 (0.88) ^b^	0.08
Height (cm)	169.6 (8.7)	167.9 (10.9)	167.0 (8.7)	0.52
Moderate activity (%)	5.05 (3.6)	4.80 (4.0)	3.78 (2.5)	0.26
Vigorous activity (%)	1.47 (2.4)	1.16 (1.97)	0.99 (1.74)	0.51
Alcohol portion (Nb)	0.52 (0.82)	0.32 (0.49)	0.23 (0.39)	0.17
Hot drink portion (Nb)	1.41 (1.09)	1.50 (1.14)	1.30 (0.74)	0.75
Whole starch portion (Nb)	1.57 (1.04)	1.41 (1.23)	1.38 (0.74)	0.54
Fruit portion (Nb)	1.26 (0.92)	1.17 (0.88)	1.53 (1.24)	0.49
Dairy product portion (Nb)	1.63 (1.00)	1.45 (1.00)	1.33 (1.04)	0.31
Refined starch portion (Nb)	1.08 (0.87)	1.47 (1.11)	1.08 (0.73)	0.12
Fatty-salty-sugary product portion (Nb)	0.85 (0.97)	1.37 (1.44)	0.49 (0.62)	0.15
Snack portion (Nb)	0.29 (0.44)	0.53 (0.72)	0.39 (0.48)	0.26
Meat-Fish-Egg portion (Nb)	1.50 (0.65)	1.21 (0.68)	1.31 (0.72)	0.18

Means within rows bearing the same or no superscript letter are not significantly different.

**Table 3 healthcare-11-01101-t003:** Correlation coefficients of Pearson between BMI and individual or behavioral characteristics in the training sample (n = 108).

Variables	r	*p*
Sedentary behavior (%)	0.51	<0.0001
Light activity (%)	−0.47	<0.0001
Food portion (Nb)	−0.21	0.03
Moderate activity (%)	−0.19	0.05
Alcohol portion (Nb)	−0.18	0.06
Fatty–salty–sugary product portion (Nb)	−0.18	0.06
Age (y)	0.17	0.08
Height (cm)	−0.17	0.08
Vegetable portion (Nb)	−0.16	0.09
Legume portion (Nb)	−0.15	0.12
Vigorous activity (%)	−0.13	0.19
Hot drink portion (Nb)	−0.12	0.20
Food balance score	−0.10	0.30
Snack portion (Nb)	0.07	0.46
Whole starch portion (Nb)	−0.07	0.47
Refined starch portion (Nb)	−0.07	0.47
Meat−Fish−Egg portion (Nb)	−0.06	0.55
Dairy product portion (Nb)	−0.06	0.55

r: value of Pearson correlation coefficient; *p*-value: probability that r ≠ 0 (n = 108).

**Table 4 healthcare-11-01101-t004:** Confusion matrix.

	Training Sample (n = 108)			Test Sample (n = 165)		
**Predicted status** **True status**	**NW**	**OW**	**OB**	**NW**	**OW**	**OB**
**NW**	**34**	2	0	**96**	5	2
**OW**	19	**9**	8	20	**13**	5
**OB**	4	10	**22**	4	4	**16**

**Table 5 healthcare-11-01101-t005:** Characteristics of true and false positives by weight status in the test sample (Mean (SD), n = 165).

True Status	Predicted Status	Size	BMI (kg/m²)	Age (y)	Light Activity (%)	Vegetable Portion (Nb)	Height (cm)
**NW**	**Total**	**103**	**21.7 (1.5)**	**32.9 (11.5)**	**22.6 (9.7)**	**1.5 (0.8)**	**170.5 (9.7)**
	NW	96	21.6 (1.5)	32.5 (11.6)	23.4 (9.4)	1.5 (0.8)	170.4 (9.9)
	OW	5	21.6 (1.1)	32.0 (6.7)	13.8 (10.4)	1.9 (1.3)	173.6 (8.1)
	OB	2	22.7 (0.1)	51.0 (2.8)	9.5 (0.7)	1.0 (0.5)	164.0 (1.4)
**OW**	**Total**	**38**	**27.4 (1.3)**	**39.6 (15.0)**	**13.8 (10.2)**	**1.6 (0.9)**	**169.5 (10.0)**
	NW	20	26.7 (1.1)	36.1 (14.9)	20.2 (10.2)	1.8 (1.1)	170.2 (10.5)
	OW	13	28.0 (1.2)	43.5 (16.5)	7.8 (1.8)	1.6 (0.6)	169.8 (10.1)
	OB	5	28.6 (1.0)	43.6 (9.5)	3.7 (0.8)	0.8 (0.5)	166.2 (8.8)
**OB**	**Total**	**24**	**35.5 (4.5)**	**42.9 (10.9)**	**8.6 (6.5)**	**1.3 (0.9)**	**165.8 (9.3)**
	NW	4	40.0 (5.5)	44.8 (8.3)	19.9 (7.7)	1.1 (0.9)	159.2 (12.2)
	OW	4	35.8 (4.3)	37.0 (9.9)	8.9 (3.5)	0.7 (0.3)	161.2 (5.6)
	OB	16	34.4 (3.8)	43.9 (11.8)	5.7 (2.5)	1.6 (0.9)	168.6 (8.5)

## Data Availability

The data presented in this study are available at https://perso.isima.fr/~lacomme/site3/healthcare/healtcare.html (accessed on 8 April 2023).

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
