# Peer review of "Weight Status Prediction Using a Neuron Network Based on Individual and Behavioral Data"

_healthcare, 2023, doi:10.3390/healthcare11081101_

Round 1

Reviewer 1 Report

This manuscript was designed to predict the weight status, such as normal, overweight, and obese of adult individuals in free-living conditions on the basis of their individual characteristics and behavioral components using Machine learning. The manuscript is interesting and I would suggest publication after the authors consider my comments.

Abstract I would suggest not initiating sentences with abbreviations (please see lines 18, 19…)

Introduction Section

Please provide the hypothesis of the study.

I think the authors could provide clinical relevance to the study.

Methodology Section

Did the author determine a sample size before starting the data collection?

Discussion Section I would recommend the author discuss the impact of age on weight status

Author Response

Reviewer 1

Remark 1. “The manuscript is interesting and I would suggest publication after the authors consider my comments.”

Answer. Thanks for your support.

Remark 2.

“Abstract I would suggest not initiating sentences with abbreviations (please see lines 18, 19…)”

 Answer. The abbreviations at the beginning of the sentence have been replaced by the whole words in Abstract, Results and Discussion.

Remark 3. Introduction Section Please provide the hypothesis of the study.

Answer. We conjectured that to be young, to have a significant physical activity and a healthy diet rich in plant products and poor in snacks, fatty-salty-sugary products could be related with normal weight status. The older the subjects, the less active they are, the less healthy their diet is, and the higher their weight status is likely to be. We add theses sentences at the end of the introduction.

Remark 4.   I think the authors could provide clinical relevance to the study.

Answer. This work is part of the prevention and fight against obesity. It contributes to the understanding of the main determinants of weight status change over time  and to the development of a mathematical model of prediction. The purpose of this work is to implement this model in a mobile application to inform people about the impact of behaviors on weight status over the years. We add theses sentences at the end of the introduction.

Remark 5. Methodology Section Did the author determine a sample size before starting the data collection?

Answer. The authors used all the data available to develop the best neuron network. In human studies, data acquisition is very expensive and time consuming.

Remark 6. Discussion Section I would recommend the author discuss the impact of age on weight status.

Answer. The authors agree to discuss the impact of age on weight status. Santos et al. (2020) studied a 10-year evolution od BMI in the same 50019 participants aged 18 to 90 years. In both periods (2007-2008 and 2017-2018) the BMI of the participants gradually increases until the age of 60 and then decreases between 60 and 90 years of age. The mean BMI values varied from 23 to 30 kg/m² and 26 to 30 kg/m², between 18 and 60 years of age during both periods. BMI was lower in younger than in older participants. Fat mass increases with age. Moreover, Han et al. (2016) showed an age-related loss of skeletal mass associated with lower muscle strength and poorer physical performance. High fat mass leads to fat infiltration into muscle. This type of infiltration contributes to decrease muscle quality and work performance (Santilli et al., 2014). Fat mass increases when energy brought by the diet becomes higher than energy expenditure associated with physical activity. Between the ages of 25 and 54 people move less and have a diminished metabolic rate (Black et al., 1990) but maintain the same energy intake (Capdevila et al., 2000).

Reviewer 2 Report

1. The number of references is not sufficient.

2. Extensive proofreading is required.

3. The sample size is small for creating a model.

4. The Shapiro–Wilk test is more appropriate method for small sample sizes (<50 samples). 

5. The manuscript did not have good quality.

Author Response

Reviewer 2

Remark 1. “Extensive editing of English language and style required”.

Answer. Gail WAGMAN provided the certificate of proofreading below:

MANUSCRIPT TITLE:

Weight status prediction using a neuron network based on individual and behavioral data

AUTHORS:

Sylvie Rousset, Aymeric Angelo, Toufik Hamadouche, Philippe Lacomme

DATE ISSUED:

March 3, 2023

                                               CERTIFICATE OF PROOFREADING

This is to certify that I, the undersigned, Gail WAGMAN, a native English speaker and professional translator and interpreter, have edited the manuscript listed above for proper English language, grammar, punctuation, spelling and overall style.

Gail Wagman

Sauve, March 3, 2023

The other reviewers rated the quality of English language as: “English language and style are fine”.

Remark 2. “The number of references is not sufficient.”

Answer. Four new references were added.

References added:

Han, D.S., Chang, K.V., Li, C.M., Lin, Y.H., Kao, T.W., Tsai KS, et al. Skeletal muscle mass adjusted by height correlated better with muscular function than that adjusted by body weight in defining sarcopenia. Sci Rep 2016, 6:19457.

Santilli, V., Bernetti, A., Mangone, M., Paoloni, M. Clinical definition of sarcopenia. Clin Cases Miner Bone Matab 2014, 11:177-180.

Black, A.E., Coward, W.A., Cole, T.J., Prentice, A.M. Human energy expenditure in affluent societies: an analysis of 574 doubly-labelled water measurements. Eur J Clin Nutr 1996, 50: 72-92.

Capdevila, F., Llop, D., Guillen, N., Luque, V., Perez, S., Selles, V., et al. Food intake, dietary habits and nutritional status of the population of Reus (Catalonia, Spain) (X): evolution of the diet and macronutrients contribution to energy intake (1983-1999), by age and sex. Medicina Clinica 2000,115(1):7-14.

Remark 3. “Extensive proofreading is required.”

Answer. The proofreading by an English speaker was done.

Remark 4. “The sample size is small for creating a model.”

Answer. It is true and this limitation is mentioned at the end of the discussion. The authors used all the existing data available to develop the best neuron network. In human studies, data acquisition is very expensive and time consuming. With this small sample, the results provided by the neural network were not so bad.  Moreover the reasons of confusion of the neural network  are given, which gives interesting tracks of improvement. This is why the authors wanted to publish them. In human health research we go step by step and our ambition is to improve the results at each new step.

Remark 5.The Shapiro–Wilk test is more appropriate method for small sample sizes (<50 samples).

Answer. In the SAS help and documentation it is written that the Shapiro-Wilk W test is computed only when the number of observations (n) is less than 2,000, while computation of the Kolmogorov-Smirnov test statistic requires at least 2,000 observations.

Remark 6. The manuscript did not have good quality.

Answer. Very few articles have used neural networks to classify weight statuses according to behavior. This is an original idea that deserves to be presented.

Reviewer 3 Report

The paper is to provide a new predictive method for current and future weight status estimation based on individual and behavioral characteristics (sex, age, height, physical activity and number of food portions). As the author said, the accuracy was not high. To increase the accuracy of the classification, a greater number of data are needed, while it is easy to do that. In addition, more variables should be selected to improve the model's prediction. Therefore, at the present time, this paper is not recommended to be published in the current format. 

Author Response

Reviewer 3

Remark 1.

The paper is to provide a new predictive method for current and future weight status estimation based on individual and behavioral characteristics (sex, age, height, physical activity and number of food portions). As the author said, the accuracy was not high. To increase the accuracy of the classification, a greater number of data are needed, while it is easy to do that. In addition, more variables should be selected to improve the model's prediction. Therefore, at the present time, this paper is not recommended to be published in the current format.

Anwser 1.

It is true that the data are few and this limitation is mentioned at the end of the discussion. The authors used all the existing data available to develop the best neuron network. it is inaccurate to say that it is easy to get more data.  In human studies, data acquisition is very expensive and time consuming.  We tested models with three, four, five, six, seven, eight or even all the variables. Models with more than four variables did not give better accuracy. The data treatment is also a long process. With this small sample, the results provided by the neural network were not so bad. Moreover the reasons of confusion of the neural network are given, which gives interesting tracks of improvement. This is why the authors wanted to publish them. In human health research we go step by step and our ambition is to improve the results at each new step.

Reviewer 4 Report

1. Was this research approved by the Ethics Committee?

2. Why were the subjects in this study classified into three groups (normal, overweight and obese) but not four groups (underweight, normal, overweight and obese)?

3. Is there any factors affecting the validity of this model? It shoul be focused in the Discussion.

Author Response

Reviewer 4

Remark 1. “Was this research approved by the Ethics Committee?

Answer. This observational study was conducted according to the guidelines laid down in the Declaration of Helsinki and the French legislation for the collection of anonymous human data. Since the existing data are completely anonymous and aggregated, the opinion of the Committee for the Protection of Individuals is not required.

Remark 2. “Why were the subjects in this study classified into three groups (normal, overweight and obese) but not four groups (underweight, normal, overweight and obese)?

Answer. Three groups were studied because there were too few underweight people to constitute a fourth group.

Remark 3. “Is there any factors affecting the validity of this model? It shoul be focused in the Discussion.“

Answer. Several factors may affect the validity of the model and they were presented in the discussion:

  • The low number of data and variability in existing data resulted in a moderate accuracy. A higher number of data (especially in overweight volunteers) could improve the accuracy of the neural network.
  • Weight and height were auto reported. Weight could be underestimated especially in volunteers whom BMI was based on cut-off values (between normal weight and overweight, or between overweight and obese). Measurement of weight and height by a health professional could improve weight status ranking.
  • Normal weight status is not the same as healthy status. Knowing metabolic status in addition to BMI would allow for a more refined ranking of volunteers and may be better linked to physical activity behavior and food choices.

Round 2

Reviewer 2 Report

The manuscript did not meet the criteria for developing a model. It does not have potential to be accepted.

Author Response

Reviewer 2

Remark .The manuscript did not meet the criteria for developing a model. It does not have potential to be accepted

Answer. The criteria for developing a model highly depends on the data available, the objective and the previous published ones. Here we introduce a new efficient model that meet the constrant we have in real life proving that the prediction highly depends on a very low number of inputs. Our contribution consist in providing that there is a trend that should push us into considering the very specific correlation we found.

Reviewer 4 Report

1. The application number provided by the Ethics Committee is required. 

2. Although there were too few underweight people, it only can be called reasonable to divide all participants into four groups according to the WHO. Thus, the authors should divide all people into four groups and predict weight status by using the neuron network based on individual and behavioral data, which can be added in the supplementary materials. Then, please compare all these results and discuss the differences. 

Author Response

Reviewer 4

Remark 1.The application number provided by the Ethics Committee is required.

Answer. In the particular case of statistical treatments carried out on existing completely anonymous and aggregated data, the agreement of an ethics committee is not required in France. The findings shown in this manuscript are the result of statistical processing of existing data. The description of the material and method could be confusing. It has been revised to remove ambiguity.

Remark 2. “Although there were too few underweight people, it only can be called reasonable to divide all participants into four groups according to the WHO. Thus, the authors should divide all people into four groups and predict weight status by using the neuron network based on individual and behavioral data, which can be added in the supplementary materials. Then, please compare all these results and discuss the differences.” 

Answer. In the Esteban study (2015), thinness (BMI < 18.5 kg/m2) is observed in 2.4% of the French adult population aged between 18 and 74 years. This percentage is very low compared to that of normalweight (48.5%), overweight (31.9%) and obese people (17.2%). These findings explain why few underweight individuals are present in our database. A very large number of subjects would have to be surveyed to have at least 80 to 100 underweight subjects.  Moreover, the underweight status can have different origins: constitutional, related to a disease or to age, or to dietary restriction. It is a heterogeneous category of people with various behaviors. To be studied, this category needs a lot of data and clarifications on the origin of the thinness. Only 33 volunteers are underweight in our database. 22 datasets were used for training but only 11 were available for the test. Eight out of 11 were classied as normal weight by the neural network. The three other volunteers were classified as underweight. At the present time, the authors have preferred to remove the underweight subjects from the statistical study due to lack of data and information about their thinness.